# The impact of the UK soft drink industry levy on the soft drink marketplace, 2017–2020: An interrupted time series analysis with comparator series

Madison Luick[1], Lauren K. Bandy[1]*, Richard Harrington[1], Jayalakshmi Vijayan[1], Jean Adams[2], Steven Cummins[3], Mike Rayner[4], Nina Rogers[2], Harry Rutter[5], Richard Smith[6], Martin White[2], Peter Scarborough[1]

1 University of Oxford, Nuffield Department of Primary Care Health Sciences, Oxford, United Kingdom, 2 MRC Epidemiology Unit, Institute of Metabolic Science, University of Cambridge School of Clinical Medicine, Cambridge, United Kingdom, 3 Population Health Innovation Lab, Department of Public Health, Environments & Society, London School of Hygiene & Tropical Medicine, London, United Kingdom, 4 University of Oxford, Nuffield Department of Population Health, Oxford, United Kingdom, 5 Department of Social and Policy Sciences, University of Bath, Bath, United Kingdom, 6 Vice Chancellor's Office, University of Exeter, Exeter, United Kingdom

* lauren.bandy@phc.ox.ac.uk

**Data Availability Statement:** All relevant data can be accessed using the following information: URL

## Abstract

### Background

In April 2018, the UK government implemented a levy on soft drinks importers and manufacturers, tiered according to the amount of sugar in drinks. The stated aim was to encourage manufacturers to reduce sugar and portion sizes. Previous evidence suggests that the policy has been successful in reducing sugar in drinks in the short-term since implementation, but their sustained effects have not been explored. This study aimed to assess the impact of the soft drink industry levy (SDIL) on sugar levels, price, portion size and use of non-sugar sweeteners in the medium-term.

### Methods and findings

Product data from 30 November 2017 to 14 March 2020 from one major UK retail supermarket were analysed (112,452 observations, 126 weekly time points). We used interrupted time series analysis, to assess the impact of the soft-drink industry levy (SDIL) on levy-eligible soft drinks, with exempt drinks (i.e. 100% fruit juices, milks, flavoured milks) acting as a comparator series. At the point of implementation of the SDIL (April 2018) there was a step change in the proportion of eligible drinks with sugar content below the SDIL levy threshold (5g per 100ml) (+0.08, 95%CI: +0.04, +0.12), with a similar sized decrease in the proportion in the highest levy category (> = 8g sugar per 100ml) (-0.06, 95%CI: -0.10, -0.03). Between April 2018 and March 2020, the proportion of eligible drinks below the SDIL levy threshold continued to gradually increase (p = 0.003), while those in the highest levy category decreased (p = 0.007). There was a step change in price of eligible drinks in the higher levy category at the point of implementation of +£0.049 (95%CI: +£0.034, +£0.065) per 100mL

= http://dx.doi.org/10.5287/ora-7jvyok0o8 DOI = 10.5287/ora-7jvyok0o8.

**Funding:** NR, MW and JA were supported by the Medical Research Council (grant Nos MC_UU_00006/7). PS is supported by the Oxford Health Biomedical Research Centre. ML and LB were funded by the NIHR Applied Research Collaboration (ARC) Oxford and Thames Valley. This project was funded by the NIHR Public Health Research programme (grant Nos 16/49/01 and 16/130/01). The views expressed are those of the authors and not necessarily those of the National Health Service, the NIHR, or the Department of Health and Social Care, UK. The funders had no role in study design, data collection and analysis, decision to publish, or preparation of the manuscript. For the purpose of Open Access, the author has applied a CC BY public copyright licence to any Author Accepted Manuscript version arising from this submission.

**Competing interests:** The authors have declared that no competing interests exist.

(for comparison, the levy is set at £0.024 per 100mL for this group). Trends in price for the high levy category were not altered by the SDIL. In the no levy category, there was a step change in price at the implementation (+£0.012 per 100mL, 95%CI: +£0.008, +£0.023), followed by a second step change in October 2018 (-£0.018p per 100mL, 95%CI: -£0.033, -£0.001p). The volume of products in the higher levy group decreased at the time of the implementation (-305mL on average including multipacks, 95%CI: -511, -99). The change in trend for the product volume of drinks in the higher levy group between April 2018 and March 2020 was in the increasing direction (+704mL per year, 95%CI: -95, 1504), but it did not meet our threshold for statistical significance (p = 0.084). There were no changes observed in the volume of lower levy drinks or no levy drinks. There was a step change in the proportion of drinks with non-sugar sweeteners at the implementation of the SDIL (+0.04, 95%CI: +0.02, +0.06).

## Conclusion

These results suggest that the SDIL was successful in [1] producing reductions in sugar levels that were maintained over the medium term up to March 2020 and [2] a reduction in product volume for higher tier drinks that may be diminishing over time. Our results also show that the SDIL was associated with a maintained price differential between high and low sugar drinks.

## Introduction

Consumption of sugar-sweetened beverages (SSBs) has been associated with an increased risk of overweight and obesity, poor oral health, diabetes, and heart disease [1–3]. To limit the harmful impacts of SSBs on health, the World Health Organization (WHO) has suggested that decisive action needs to be taken, including the implementation of taxes on SSBs and restrictions on their marketing [4]. As of March 2022, more than eighty-five countries globally have proposed or implemented some form of SSB tax [5].

In 2016, the UK Government announced it would be introducing a tax on SSBs (the Soft Drinks Industry Levy–SDIL), to be implemented in April 2018 [6]. The levy was targeted at importers and manufacturers of SSBs and has two tiers: drinks with more than 8g of sugar per 100 mL are levied at £0.024 per 100mL and those with 5g to 8g of sugar per 100mL at £0.018 per 100mL; those with less than 5g of sugar per 100mL are not levied [7]. The delay between announcing the levy and its implementation allowed time for manufacturers to reformulate soft drinks to avoid the levy. Exempt from the levy, irrespective of sugar content, are drinks which contain at least 75% milk or a milk alternative, 100% fruit juice, more than 1.2% alcohol by volume, or are produced by a manufacturer that produces less than 1 million litres per year [8].

Previous research on the impact of the SDIL has found no evidence of a reduction in consumer purchases of soft drinks by volume [9]. However, in response to the announcement of the SDIL, soft drink manufacturers have been shown to have reformulated their products to have less sugar in order to avoid the highest levy, and also to have increased the price of products in the highest levy category [10]. The introduction of the levy has also been associated with a decrease in rates of obesity prevalence in English girls in their final year of primary school [11] and hospital admissions for extraction of carious teeth in children [12].

Previous research that showed evidence of reformulation in the lead up to SDIL implementation only had access to a little under one year of data following the implementation [10]. A

recent systematic review identified that initial analyses of tiered taxes have shown they are associated with reformulation and a reduction of sugar content, but it also identified the need to continue analysing manufacturer and purchasing responses, as these may change over time [13]. Long-term trends in purchasing behaviour have been explored e.g. in Mexico, where a tax was added to SSBs in 2014, there is evidence shows sustained effects through 2018 in consumer purchasing patterns [14]. However, longer term effects on manufacturer and retailer responses are less well known. Since the rates of SDIL were not linked with inflation, it is possible that the declining real size of the levy over time could influence industry behaviour. Studies that analyse the more medium- and long-term impacts of legislation such as the SDIL can help provide a better understanding of the persistence of the effects elicited with the implementation of these policies.

In this study, we examine observed medium-term manufacturer responses to SDIL up to two years after the implementation of the legislation. Our study covers a period from late November 2017 to March 2020, ending prior to any evidence of effects from COVID-19. It explores changes in grams of sugar per 100 mL, price, product volume, and use of non-sugar sweeteners in products in the two years following the SDIL implementation.

## Methods

This study did not involve any data from human participants and therefore did not require ethical approval.

### Study design

We undertook an interrupted time series analysis of levy-eligible drinks, analysing trends before and after SDIL was implemented on 6 April 2018, with a comparator series of drinks that were exempt from the legislation (i.e. 100% fruit juices, milk, flavoured milk drinks and milkshakes). Levy-eligible drinks that avoid the levy due to low sugar levels e.g. diet sodas are included in the primary analyses (8). Analyses using the comparator series were conducted independently of the main analyses but using the same regression model structure. The hypothesis was that the primary analyses would demonstrate changes in the outcome variables due to implementation of the SDIL, but analyses of the comparator series would show null results (thereby demonstrating specificity of the effect of the SDIL). Given the potential for spill over effects from SDIL, this was not run as a controlled ITS. We refer to the two sets of drinks as 'eligible drinks' (for the primary analyses) and 'exempt drinks' (for the comparator analyses).

Eligible drinks were further classified based on their sugar content. For the analyses that explored the proportion of drinks below 5g or at or above 8g sugar per 100mL, the sugar content at the specific time point of the analysis was used to classify the drink. For analyses that examined price, product volume and NSS, each drink was classified by the sugar content it had at the final time point in the dataset. For example, Irn Bru (2 Litre) reformulated from 10.3g of sugar/100mL to 4.7g of sugar/100 mL. Irn Bru (2 Litre) first appears in the dataset on 30 November 2017, and last on 14 March 2020. We used the sugar content value at 14 March 2020 to classify Irn Bru (2 Litre) in analyses for price, product volume, and use of NSS to show how they changed over time based on the final levy tier. This was done to avoid confounding of results for these analyses as drinks move between categories due to reformulation.

### Data

Data were obtained from foodDB [15], a database set up in 2017 that uses an automated web scraping tool to provide data on food and drinks available on UK supermarket websites. This

data represents all available offerings on the supermarket website but does not take account of differing sales of drinks. For this analysis, we used data from Tesco which has over a quarter of the market share in the UK, making it the leading retail supermarket [16]. The data comprised 126 time points of soft drink data from this one major UK retail supermarket chain, taken approximately one week apart for the period 30 November 2017 to 14 March 2020. Each line of data is for a drink that appeared on the supermarket website at the point of time when data were collected. Variables included date of data collection, drink name, product volume, price, sugar content (g per 100mL), and ingredients lists. For multipack items, the product volume and price were for the total product (i.e. the full pack size). So, if a multipack of four 330mL cans was purchased, the item was treated as 1320mL, and value per 100mL was calculated based on this product volume and the total pack price. Where sugar content was missing, it was imputed with the nearest available value for the same product before the missing value, or (if no earlier versions of the product were available) the nearest available value after the missing value. For 39 drinks we were unable to identify sugar content at any point in the dataset, so all data points associated with these drinks were removed (amounting to 1.04% of the dataset). The final dataset for analysis included 112,452 data points over 126 time points, with each data point representing a separate stock keeping unit (SKU) with a unique brand and size at the unique time point.

The NSS outcome variable was based on presence of at least one of the following ingredients in the ingredients list (adapted from [17]): Acesulfame K (Acesulfame K, Equal, Twinsweet, Sunett, Sweet one, E950, E962); Alitame (Alitame, E956); Aspartame (Aspartame, Equal, Instasweet, Nutrasweet, Twinsweet, E951, E962); Cyclamate (Clyclamate, E952); Monkfruit extract (Monkfruit, Luchan guo); Neotame (Neotame, Newtame, E961); Saccharin (Saccharin, Nectasweet, Sugartwin, Sweet'n low, E954); Steviol glycosides (Steviol glycosides, Enliten, Candyleaf, Rebaudioside A, Reb-a, Rebiana, Sugarleaf, Sweetleaf, Truvia, Stevia, Purevia, E960); Sucralose (Sucralose, Altern, Kaltame, Splenda, Trichloroglactosucrose, E955); Thaumatin (Thaumatin, E957); Erythritol (Erythritol, E968); Isomalt (Isomalt, E953); Lactitol (Lactitol, E966); Maltitol (Maltitol, E965); Sorbitol (Sorbitol, E420); Mannitol (Mannitol, E961); Polyglycitol (Polyglycitol, E964); Xylitol (Xylitol, E967).

The price outcome variable was calculated as £ per 100 mL based on the price and product volume data in the dataset. The price in the dataset represents the price consumers would have to pay to purchase a single item as offered for purchase, whether this was a single drink or a multipack, at that time point. The price incorporates reductions due to price promotions (e.g. 10% off) but not due to volume promotions (e.g. buy one get one free). For example, if an eight pack of 330mL cans costs £3.50, the price (£) per 100 mL is calculated using £3.50 and 2640mL, i.e. £0.13.

## Outcome measures

Our outcome measures were:

- The proportion of soft drinks with sugar content less than 5 g per 100 mL, which corresponds to those drinks which have no levy applied to them

- The proportion of soft drinks with 8 g of sugar per 100 mL or more, which relates to the level at which drinks have the higher levy applied

- The average price of soft drinks (£ per 100 mL) available

- The average volume of soft drinks (mL) available

- The proportion of soft drinks with non-sugar sweeteners (NSS) listed in their ingredients.

Items were considered by SKU, so those outcome measures which are a proportion, were the proportion of all SKU on a given date, and those that are averages were the average of all SKU on the given date.

## Statistical methods

The dataset was divided into five time segments, each approximately six months long; in some periods data collection was not quite six months–this is because: a) foodDB data collection did not start until mid-November 2017, less than six months before implementation of the SDIL; and b) data collection from foodDB was abruptly halted in March 2020 due to disruptions caused by the first COVID-19 lockdown:

- Time period 1: 30th Nov 2017 – 5th Apr 2018

- Time period 2: 6th Apr 2018 – 30th Sep 2018

- Time period 3: 1st Oct 2018 – 31st Mar 2019

- Time period 4: 1st Apr 2019 – 30th Sep 2019

- Time period 5: 1st Oct 2019 – 14th Mar 2020

We divided the interrupted time series into 6-monthly periods to mitigate the situation where a data point at the end of the time series could influence the interpretation of a change at a much earlier time point, in this case the point of SDIL implementation. This was particularly important for our analysis, where there was an imbalance of data before and after the SDIL implementation point of 6 April 2018, with only six months of data before and almost two years after this date. We applied a simple model, which allowed for a slope change at the point of implementation of the SDIL, but does not permit slope changes between the post-implementation time periods, to avoid issues caused by over-fitting the data. The data were visually assessed, and some seasonal trends were observed. A previous analysis of SDIL found evidence of a seasonal change in December [10], and our visual assessment showed periodic changes, which occurred during the summer months, so a dummy variable was included for the month of December and another dummy variable for the summer months (i.e. June, July, and August).

Interrupted time series analysis was used to analyse data that had been aggregated for each time point (proportions were used to aggregate drinks based on sugar category and presence of NSS; arithmetic means were used to aggregate data on price and product volume). The regression models allowed for level changes between each time-period and a slope change between pre-implementation and all post-implementation periods combined. All models were run with Newey-West robust standard errors (lag 7 for autocorrelation with up to 7 previous data points) to account for autocorrelation in the data. The general equation for the models is shown below.

$$y = \alpha + \beta_1 I + \beta_2 t + \beta_3 I*t + \beta_4 Post_2 + \beta_5 Post_3 + \beta_6 Post_4 + \beta_5 Summer + \beta_6 Dec + \varepsilon$$

Where:

- y is the outcome variable

- $\alpha$ is the intercept

- $\beta$ are regression coefficients

- t is time (in days)

- I is a binary variable which is 0 before the implementation of the SDIL and 1 after

- $Post_2$ is a binary variable which is 1 in time period 3 and 0 elsewhere

- $Post_3$ is a binary variable which is 1 in time period 4 and 0 elsewhere

- $Post_4$ is a binary variable which is 1 in time period 5 and 0 elsewhere

- Summer is a binary variable which is 1 in June, July and August and 0 elsewhere

- Dec is a binary variable which is 1 in December and 0 elsewhere

For the analyses of price, product volume and presence of NSS, the regression models were run separately for each levy category (i.e. no levy, lower levy, higher levy) due to a hypothesis that given the tiered nature of SDIL, if there were any change in these outcomes, manufacturers may alter price, product volume, or the use of NSS differentially based on the levy applied. This was observed in our previous study that examined the announcement of SDIL and the short-term effects of its implementation [10].

A sensitivity analysis was run for price that accounted for the lowest possible price a product could be. For example, if the product was offered as part of a "Buy 1 get 1 Free" deal, the price for the sensitivity analysis was 50% of the offering price (i.e. if £1 the price for analysis was 50p).

## Display of results

We present descriptive statistics showing median and interquartile range for each outcome variable in each time period stratified by eligible / exempt drinks. The median and IQR are used as these variables were not normally distributed, and a Wilcoxon rank sum test is conducted to explore how the values differ before and after SDIL was implemented. We produced Sankey diagrams showing how drinks changed levy category from the first time point in the dataset compared to the last, both for eligible drinks and the exempt drinks. For these diagrams, we included all drinks where observations were available both before and after the implementation of the SDIL and compared their sugar levels from the first to the last entry in the dataset. Results of the regression models are displayed in figures showing modelled fit lines (with the seasonal effects removed from the fit lines for the sake of clarity). Full regression model results (including the seasonal variables) are provided in S1 Table in S1 File.

All analyses were completed in RStudio using R version 4.1.3. Dataset development was done using Python version 3.10.

## Changes to protocol

The pre-published protocol (https://njl-admin.nihr.ac.uk/document/download/2010886) states that we would analyse the impact of the SDIL on mean sugar content of drinks—upon reflection we considered that a categorical classification of the data was more relevant to the manufacturer response to the SDIL. In the protocol, we proposed using alcoholic drinks as the comparator series; this was altered because most alcoholic drinks do not report sugar content. We originally intended the end date of the analysis to be in April 2020 –this was altered because of disruption to data collection during the first COVID-19 lockdown. We had originally planned to use data from several online supermarkets, however, due to reduced staff resources and capacities for foodDB, we could only analyse data from a single supermarket. In the protocol we proposed to stratify results by branded and own-brand status, since supermarkets seem to treat their own brand products differently, and to look at product diversification, but we have not conducted those analyses as we only have data from a single supermarket.

**Table 1. Descriptive statistics of foodDB data used in the study.** Where N is the number of observations in the dataset, and n is the number of time points.

| | Pre-implementation (30th Nov 2017 – 5th Apr 2018) | | Post-implementation (6th Apr 2018 – 14th Mar 2020) | | P-value[1] (eligible) | P-value[1] (exempt) |
|---|---|---|---|---|---|---|
| | Eligible (N = 14,297, n = 21) | Exempt (N = 5,472, n = 21) | Eligible (N = 66,424, n = 105) | Exempt (N = 26,259, n = 105) | | |
| Sugar (mean per time point) (g per 100 mL) (Median, IQR) | 3.7 (3.65–3.69) | 8.4 (8.37–8.44) | 2.8 (2.72–2.85) | 8.4 (8.32–8.41) | <0.0001 | 0.031 |
| Product volume (mean per time point) (in mL) (Median, IQR) | 1263 (1250–1267) | 1021 (1015–1024) | 1245 (1239–1290) | 990 (976–1004) | 0.146 | <0.0001 |
| Price (mean per time point) (£/100mL)* (Median, IQR) | 0.23 (0.22–0.23) | 0.23 (0.23–0.23) | 0.22 (0.21–0.23) | 0.24 (0.23–0.24) | 0.069 | 0.110 |
| Non-sugar sweeteners used in drink (proportion per time point) (Median, IQR) | 0.73 (0.72–0.73) | 0.01 (0.01–0.02) | 0.80 (0.79–0.80) | 0.01 (0.01–0.02) | <0.0001 | 0.950 |

*There were different number of observations for price. Pre-implementation: eligible (n = 14,083), exempt (n = 5,421). Post-implementation: eligible (n = 65,125), exempt (n = 25,847). Difference is due to missing data.

[1] p-values come from Wilcoxon rank sum test examining the difference between drinks pre- and post-implementation

## Results

The original dataset had 126,991 observations over 127 time points. Data capture on 20 July 2018 appeared to have failed, with only half of the observations of other time points reported, so it was removed from the dataset. A few non-food items (e.g. glassware), powdered drinks (e.g. powdered milk or lemonade), or milk alternatives were also erroneously included in the dataset, and were removed. After removing products with no data on sugar or product volume, the resulting dataset contained 112,452 observations over 126 time points.

Most observations analysed in the study were levy eligible drinks (80,721 of 112,452), with most of these falling in the no levy category (69,570 of 80,721). A few products were missing price data (1,976; 1.75%). These products were not removed, but price analyses consequently have a reduced number of observations. Full descriptive statistics are found in Table 1.

### Reformulation

Sankey diagrams displaying how the eligible (a) and exempt (b) drinks changed levy categories are shown in Fig 1. Of the 955 drinks with unique SKU that appeared in the dataset both before

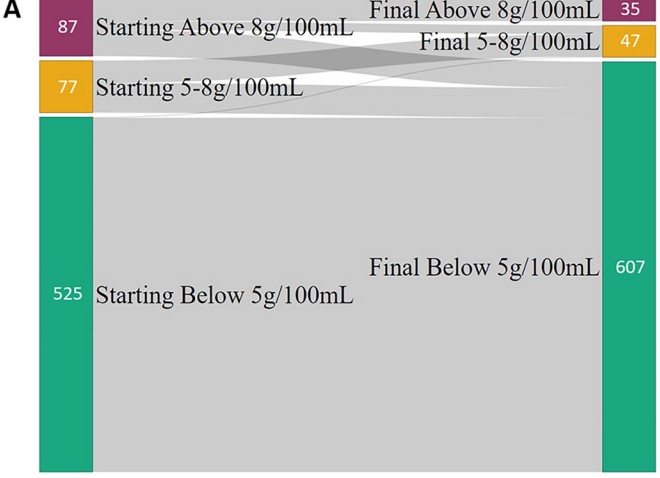
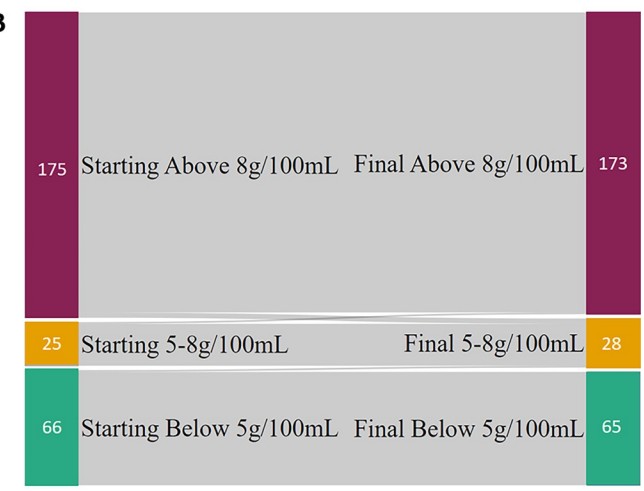

**Fig 1.** Sankey diagram for the movement of eligible (a) and exempt (b) drinks between levy group classifications over time. Starting and final sugar level are determined by the first and last time a product appears within the dataset.

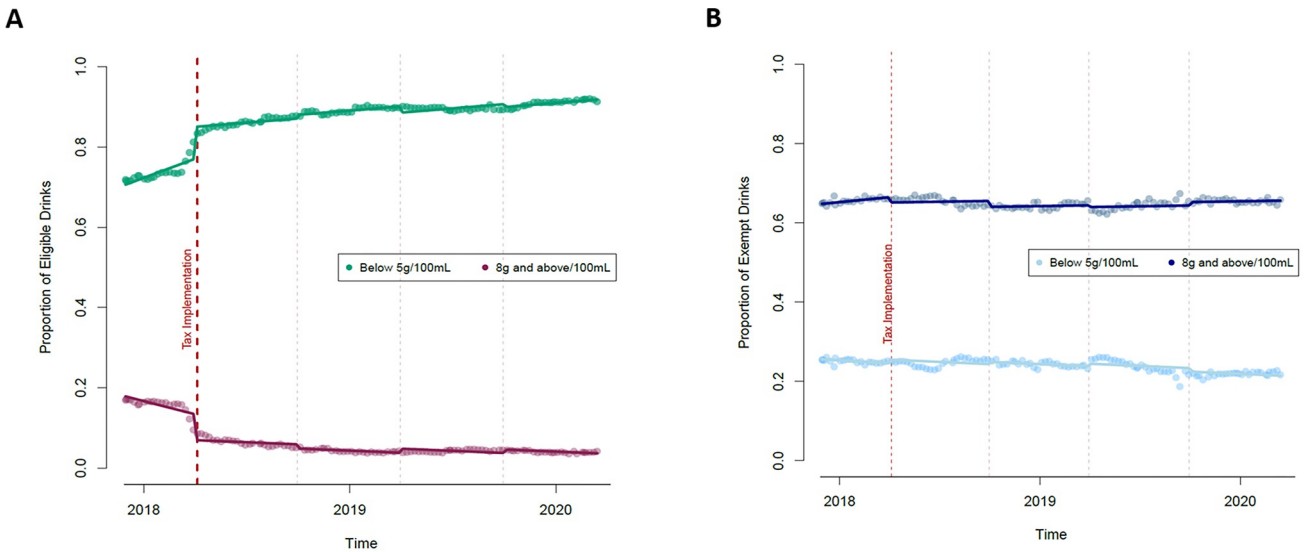

**Fig 2.** Proportion of eligible drinks (a) and exempt drinks (b) with at or above 8g/100mL or below 5g/100mL of sugar over time.

and after SDIL implementation, we observed 87 eligible drinks in the higher levy group, of which 13 (15%) moved to the lower levy group and 39 (45%) moved to the no levy group by the end of the dataset (Fig 1). In contrast, only three (2%) of the 175 exempt drinks that began with over 8g of sugar/100mL reduced their sugar content below this level.

Fig 2 shows changes in the proportion of levy eligible (a) and exempt (b) drinks with sugar less than 5g per 100mL and greater than 8g per 100mL over time. At the time of implementation, on 6 April 2018, the proportion of eligible drinks with less than 5 g of sugar per 100 mL increased in a step change (+0.08, 95%CI: +0.04, +0.12), with a similar-sized decrease in the proportion with 8g of sugar or more per 100 mL (-0.06, 95%CI: -0.10, -0.03). From 6 April 2018 to 14 March 2020, there was a slope change indicating slowing in pre-implementation trends which left a slight downward trend in the proportion of eligible drinks with 8g of sugar or more per 100 mL (p = 0.007), and a corresponding slight upward trend for the proportion of drinks with under 5 g of sugar per 100 mL (p = 0.003). No step changes were observed in the exempt drinks, but there was a small slope change in the proportion of drinks with 8g or more of sugar per 100 mL (p = 0.006) after implementation of the SDIL from 6 April 2018 to 14 March 2020.

## Price

There was a step change in the price of eligible drinks in the higher levy category on 6 April 2018, at the point of implementation, by +£0.049 (95%CI: +£0.034, +£0.065) per 100mL. Since the levy for this category is £0.024 per 100mL, this represents a pass-through rate of 204% (Fig 3). After this, prices continued to rise in this group at the same rate as prior to the implementation throughout the post implementation period, ending on 14 March 2020 (i.e. there was no slope change or step changes after implementation).

Prices in the lower levy category were erratic across the period of data collection. We observed large step changes in prices between the different 6-month time periods, but due to high variability for this category these jumps could be due to chance (e.g. p values for all time period parameters were greater than 0.05, see S1 File (S1 Table)). After 6 April 2018, there was a slope change indicating an increasing trend in the price of products in the lower levy

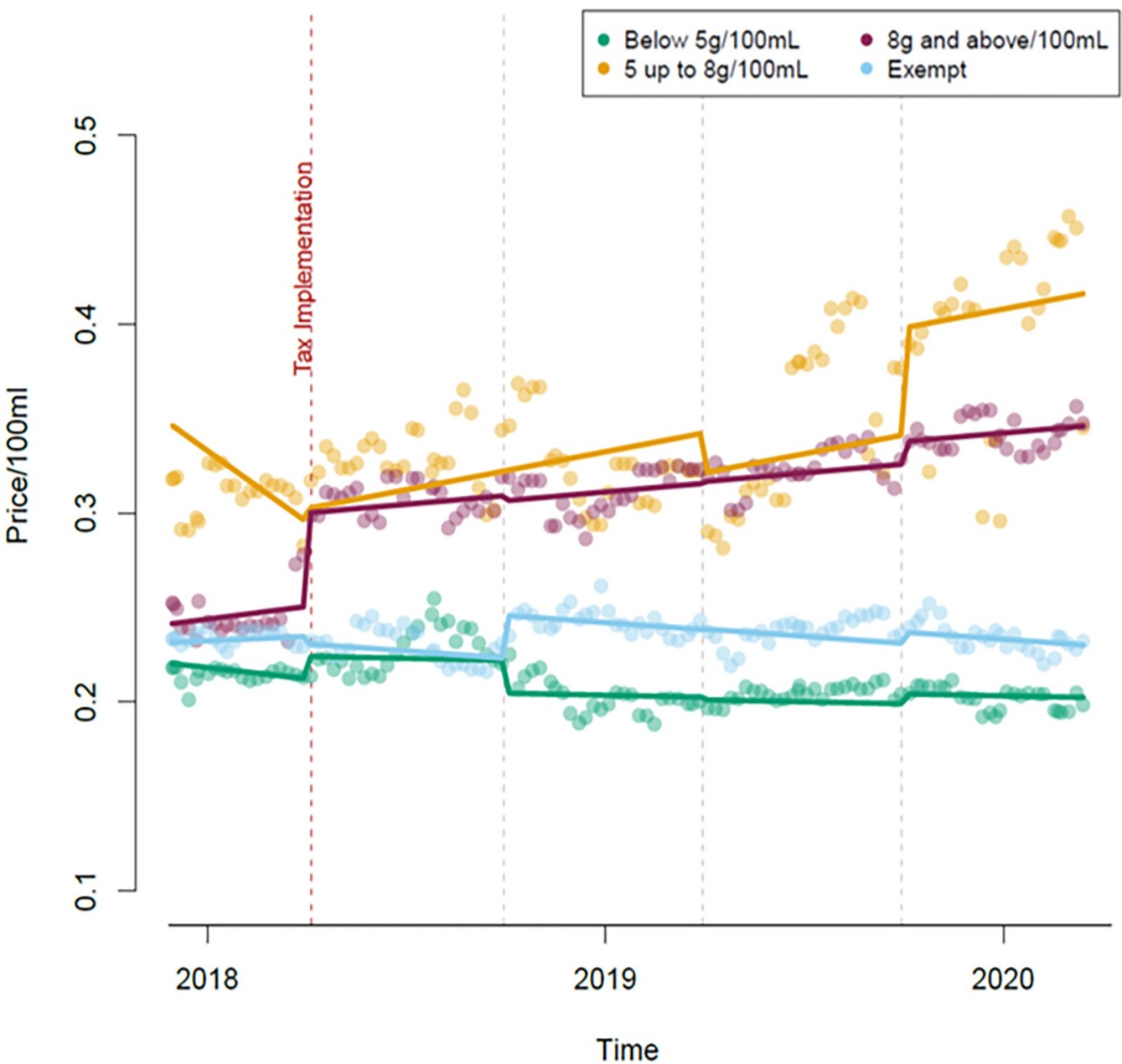

**Fig 3. Price per 100mL of drinks, classified as either control or the levy group they were in last in the dataset.**

category (+£0.0005 per 100mL per day, 95%CI: +£0.0002, +0.0008), through to 14 March 2020.

Eligible drinks with less than 5g sugar per 100mL were observed to have a small step change in price at the time of the implementation (+£0.012, 95%CI: +£0.008, +£0.023), followed by a similar decrease in price six months later (-£0.018, 95%CI: -£0.033, -£0.001). No slope change was observed. In the exempt series, no step change was observed at the time of the intervention, but prices rose six months later (+£0.023, 95%CI: +£0.006, +£0.038). There was no slope change in the comparator series.

The sensitivity analysis run which used the lowest possible price a product could have been purchased for (i.e. assumed multi-buy deals were utilised), found a similar pattern of results (see S1 File (S2 Table)).

## Volume

There was a step change in the volume of products in the higher levy group at the time of the implementation on 6 April 2018 (-305mL, 95%CI: -511, -99) (Fig 4). There was weak evidence (p = 0.084) of a slope change after implementation of the SDIL, producing an upward trend in the product volume of drinks in the higher levy group from 6 April 2018 to 14 March 2020 (+704mL per year, 95%CI: -95, 1504). Six months after implementation, there was a step change in the volume of products in the no levy group (-30mL, 95%CI: -49, -12). There were

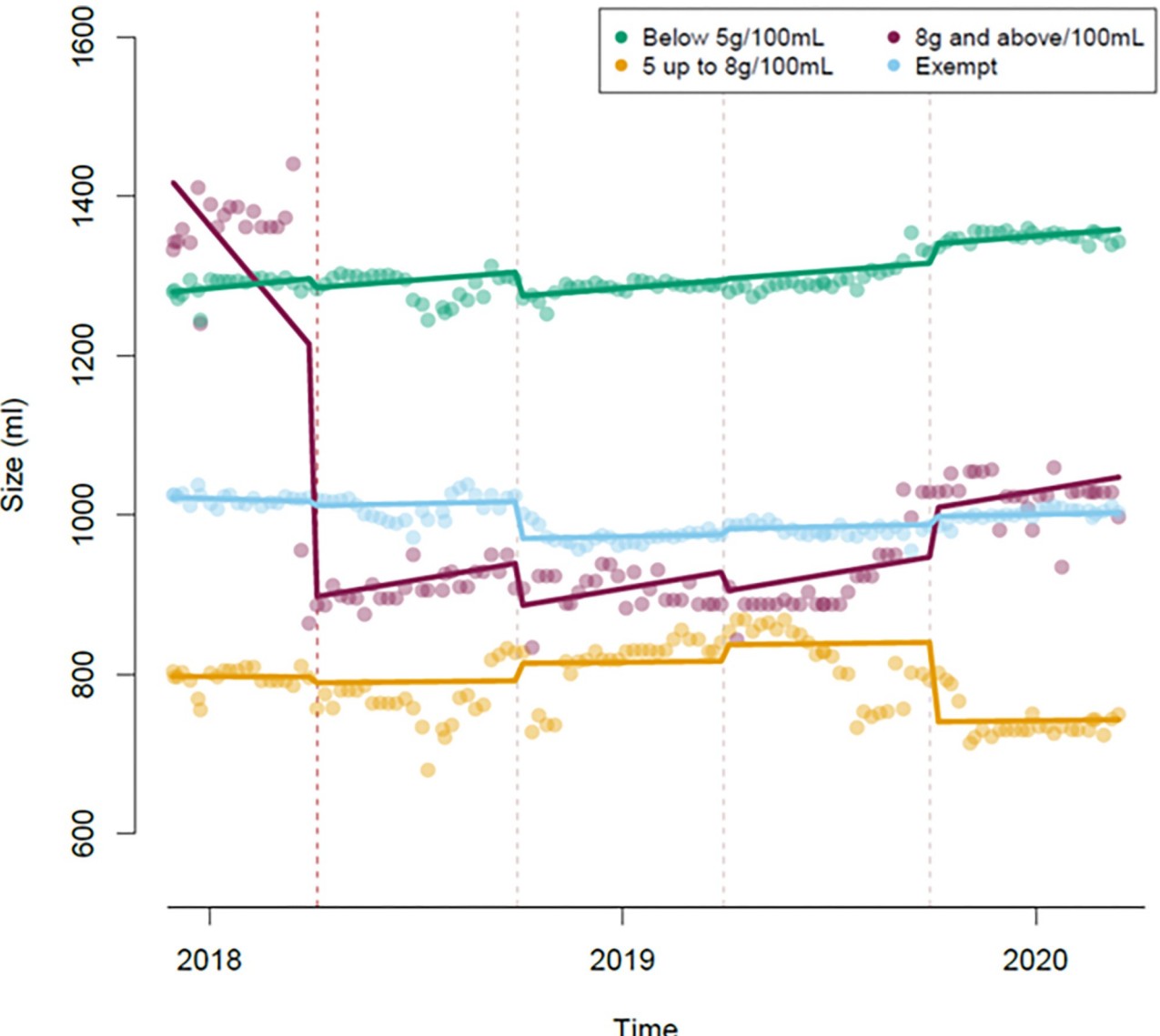

**Fig 4. Volume in mL of drinks, classified as either control or the levy group in which they were in last observed in the dataset.**

no other step or slope changes observed in the volume of no levy drinks and no significant changes observed in the volume of lower levy drinks.

There were two step changes in the product volume of exempt drinks, the first six months after the implementation (-46mL, 95%CI: -65, -28) and then at one year after the implementation (-39mL, 95%CI: -63, -15) (both in comparison with the first period after the implementation of the levy, April 2018 –October 2018).

### Non-sugar sweeteners

Fig 5 shows a step change in the proportion of eligible drinks using non-sugar sweeteners at the time of the implementation on 6 April 2018 (+0.04, 95%CI: +0.02, +0.06). Exempt drinks were observed to have a step change in proportion of drinks using non-sugar sweeteners on 1 April 2019, approximately one year after the implementation (-0.01, 95%CI: -0.02, -0.00).

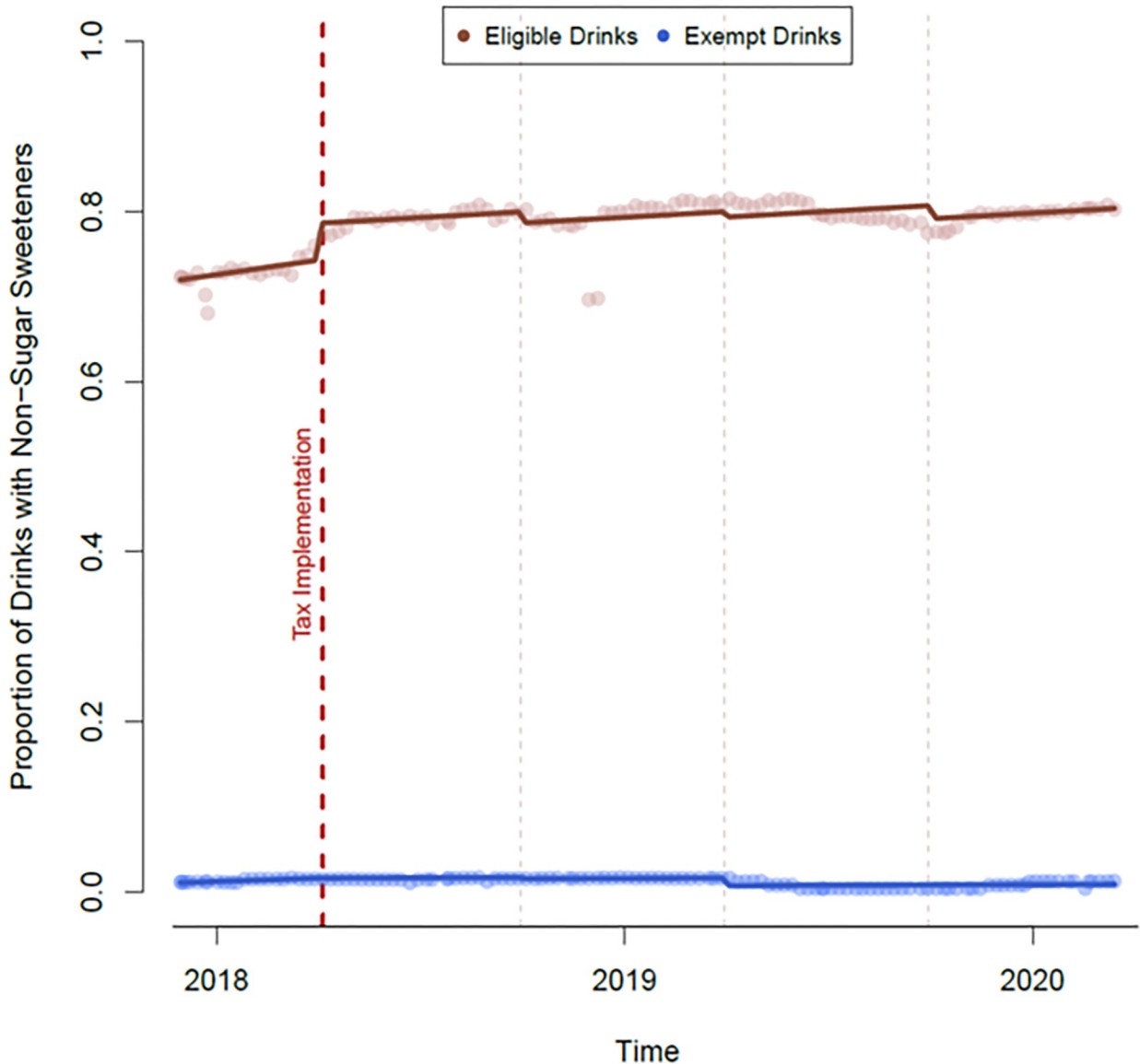

**Fig 5. Proportion of non-sugar sweeteners used in eligible and exempt drinks over time.**

## Discussion

The analyses presented here suggest that the implementation of the SDIL was associated with changes in the formulation, price and product volume of soft drinks available in the UK marketplace in expected directions. These changes were observed primarily as step changes at or shortly after the SDIL implementation. The observed changes persisted over the medium-term (up to two years after the implementation of the policy), possibly suggesting that continuation of the SDIL as a levy has led to the maintenance of these step changes. There is weak evidence that the reduction in product volume of higher levy drinks at the time of implementation may be reversing, with average product volumes moving towards levels seen before the levy. This requires further longer-term monitoring.

Differences in the formulation of soft drinks (reductions in sugar levels and increases in the use of NSS) observed here were small. The SDIL was announced in March 2016 and implemented two years later–previous analyses have shown that most of the reformulation of soft drinks happened earlier than the current data collection for this paper started (10). Our analyses of price and product volume suggest that the levy has been fully passed onto consumers in the form of consistently higher prices of soft drinks with more than 8g sugar per 100mL (the higher levy category). However, after the implementation of the levy these price rises were somewhat obscured by reductions in the product volume of higher levy drinks (i.e. 'shrinkflation') and by temporary raised prices of drinks in the 'no levy' category. Further research could explore the motivation for manufacturers' adjustments of products, lending a better understanding to the mechanistic action and responses of policies such as SDIL.

We observed erratic changes of price and volume of drinks in the lower levy category. This is a niche set of drinks largely consisting of mixers for alcoholic drinks or soft drinks aimed primarily at the adult market. These drinks have little in common with either the no levy or high levy drinks. This can be observed in the Sankey diagram, which shows that most of the drinks that left the higher levy group proceeded straight to the no levy group, rather than taking the smaller step to the lower levy category. The lower levy drinks tend to be more expensive than other soft drinks (both eligible and exempt) and available in smaller containers. There also appeared to be significant seasonal effects for this category of drinks.

### Strengths and limitations

This analysis is based on data from up to two years after the implementation date of the SDIL to assess the medium-term impacts of the policy. It utilises an expansive dataset, which provides data on offered beverage products for the UK's largest retail supermarket. However, there were some limitations. First, there was a lack of available sales data, meaning that no sales weighting could be applied in the analysis. Without sales weighting, our analysis focuses on what was available from one supermarket on their online web sales platform, rather than what was actually purchased or consumed or even what customers may see in a physical store. The data can show whether or how manufacturers changed their products in response to the legislation, but it does not provide insights into the consumption of soft drinks, or the effects of changes in sugar levels, price, or product volume at the population-level. Although purchasing data was not analysed in this study, it has been analysed previously, finding that the volume of soft drinks purchased did not decrease when the SDIL was implemented, but the amount of sugar purchased from drinks decreased, with an estimated 30% reduction [9].

We used data from the largest retail supermarket in the UK, accounting for a little over a quarter of all UK grocery sales [18]. Own-label products represent a significant volume of soft drink sales and have been shown to have differential levels of reformulation in the past [10], but with data from just one retailer, any analyses on own-label products would not have been

representative of the UK marketplace. The data collected using the foodDB platform reflects the online supermarket setting and offerings, not necessarily the in-store environment, particularly for small stores, which may stock fewer products. Nevertheless, online grocery shopping at this retailer accounts for 13% of supermarket sales, and products available in the online supermarket represent an estimated 85% of available products in stores [16, 19]. The exempt series was included as a group of drinks that is not subject to the legislation, and so was less likely to have changed in response to SDIL implementation, but with the potential for spill over effects. However, natural fluctuations in the market are inevitable as a result of discontinuation or introduction of products. Some of the drinks in the exempt series may also be potential substitute drinks for the eligible soft drinks, meaning that this group is not an ideal control series. For this reason, we analysed the data as a comparator series. Interrupted time series analyses are vulnerable to temporal confounding–that is, the results displayed here could be due to other causes that happen at a similar time.

WHO has suggested there are potential health effects of non-sugar sweeteners (NSS) [20]. Analysis of NSS was based on a binary outcome: whether or not a product contained any NSS. Therefore, our analyses only demonstrate that the implementation of the SDIL resulted in a small increase in the proportion of drinks that contained NSS, but cannot tell us about how the SDIL influenced the *amount* of NSS used within products, as this information is not available through foodDB.

## Comparison with other studies

Our study adds to existing knowledge of SSB tiered taxes and their medium-term impacts. A recent systematic review reported that an estimated 82% of the cost of SSB taxes are passed on to consumers; however, it also identified a lack of studies looking beyond the short-term impacts of these taxes [13]. In Mexico, where a soft drink tax has been implemented since 2014, by exploring the medium-term effects of the SSB tax, it was found that purchases of soft drinks decreased by an even greater amount two years following the tax implementation than in the year following, suggesting the effects of SSB taxes may cumulate over time [21]. Our study helps fill this gap in knowledge, by focusing on the medium-term impacts of the SDIL on the soft drink market. Previously, other studies on tiered SSB taxes have found similar short-term trends as in the UK. Both South Africa and Portugal, where tiered SSB taxes were implemented, had observed reductions in the sugar concentration in beverages and a price increase [22–24]. In South Africa, a similar price increase was observed for beverages with lower and higher sugar content, despite some of the drinks with lower sugar content having an effective taxation rate of zero [24], further suggesting it may not be directly correlated with the levy, but was instead the result of manufacturer manipulation. Some countries, such as France, have even redesigned an existing SSB tax to apply a tiered structure, with the aim of improving its impact and encouraging reformulation, although evaluation is still needed [25]. Our study contributes to the understanding of how a tiered SSB tax may impact soft drinks, but more research should be done in other contexts with tiered SSB taxes, both on the consumer and the manufacturer, to consider if the same effects are found or if they persist in the long-term.

## Implications

It has previously been observed that the announcement of the SDIL was associated with reductions in sugar content of drinks prior to implementation [10]. Here we show that those reductions were maintained over the medium-term following implementation, further supporting the potential public health benefits of the tiered design of the levy. It also suggests a reduction in product volume of higher sugar drinks has generally been maintained over the medium-

term, with weak evidence of a slow increase in product volume of higher levy drinks after the implementation of the levy. This supports one of the policy objectives of the SDIL to reduce the product volume of higher sugar drinks [6].

Since its announcement in March 2016, the SDIL has never been far from the headlines in the UK. It has prominent opponents (e.g. multiple suggestions from incoming Prime Ministers that the levy may be scrapped [26, 27]) and defenders (e.g. the National Food Strategy called for the levy to be extended into a sugar tax on all food and drink [28]. Our analyses suggest that the SDIL elicited its intended effect, and through the continued application of the SDIL, its effects had not diminished up until March 2020. However, since our dataset only extends to March 2020, it does not include the recent sharp increase in food prices observed as a result of the cost-of-living crisis in the UK. Unlike UK tobacco taxes, the SDIL was not index-linked, meaning that the levy does not change as retail prices change. As a result, the effective real-terms value of the levy has not kept up with inflation, and each year it will become lower as a proportion of price, and therefore may become less effective. While this did not have a material impact on the findings of this study since inflation remained low during the study period, further analyses are needed on more recent data to measure the impact of the SDIL over the longer term to observe whether this lack of index-linking results in diminishing benefits of the SDIL.

These results could inform future public health policy in the UK in the following ways. First, around three quarters of the drinks in our comparator dataset had more than 5g sugar per 100mL. These exempt drinks include flavoured milks with high amounts of added sugar. The SDIL could be extended to incorporate these drinks as a prompt towards either reformulation or reduction in product volume. Second, it is not clear whether the reduction in product volume observed in the higher levy category of drinks has been a temporary phenomenon. If longer term follow-up confirms the drift of product volume back towards pre-implementation levels, then other policy options to reduce product volume should be considered including index-linking of the SDIL and mandatory maximum product volumes.

## Supporting information

**S1 File. Full regression outputs and sensitivity analysis.**
(PDF)

## Author Contributions

**Conceptualization:** Richard Harrington, Jean Adams, Steven Cummins, Mike Rayner, Harry Rutter, Richard Smith, Martin White, Peter Scarborough.

**Data curation:** Richard Harrington, Jayalakshmi Vijayan, Peter Scarborough.

**Formal analysis:** Madison Luick, Lauren K. Bandy, Jayalakshmi Vijayan.

**Funding acquisition:** Jean Adams, Steven Cummins, Mike Rayner, Harry Rutter, Richard Smith, Martin White, Peter Scarborough.

**Investigation:** Lauren K. Bandy, Richard Harrington.

**Methodology:** Richard Harrington, Jean Adams, Mike Rayner, Martin White, Peter Scarborough.

**Visualization:** Madison Luick.

**Writing – original draft:** Madison Luick, Lauren K. Bandy, Steven Cummins.

**Writing – review & editing:** Madison Luick, Lauren K. Bandy, Richard Harrington, Jaya-lakshmi Vijayan, Jean Adams, Mike Rayner, Nina Rogers, Harry Rutter, Richard Smith, Martin White, Peter Scarborough.

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
