## [Decision Letter · Decision Letter 0]

21 Dec 2023

PONE-D-23-34193The impact of the UK Soft Drink Industry Levy on the soft drink marketplace, 2017 – 2020: an interrupted time series analysis with comparator seriesPLOS ONE

Dear Dr. Bandy,

Thank you for submitting your manuscript to PLOS ONE. After careful consideration, we feel that it has merit but does not fully meet PLOS ONE’s publication criteria as it currently stands. Therefore, we invite you to submit a revised version of the manuscript that addresses the points raised during the review process.

We have now received two reviews from experts in your field. While they expressed strong appreciation of your work, they have several fundamental concerns regarding analysis and presentation of information about the analysis. We hope that you will find the points made by the reviewers supportive for making your manuscript clearer, more convincing and valid regarding the statistical analysis and presentation of data. Please see reviewers concerns below and we look forward to the altered manuscript. 

We look forward to receiving your revised manuscript.

Kind regards,

Hans-Peter Kubis, PD. Dr. rer. nat.

Academic Editor

PLOS ONE

3. In the online submission form, you indicated that [The data used for this analysis are extracted from the foodDB project. These data are collected automatically from online supermarkets under exemption of copyright law for non-commercial research. They cannot be openly shared as commercial use of the data would breach copyright law. An anonymised version of the dataset (with product names redacted) will be made freely available on the Oxford University Research Archive upon acceptance of the manuscript. An unredacted version of the dataset is available upon request from the authors and completion of a licensed agreement to restrict use to non-commercial research by contacting Prof Pete Scarborough, peter.scarborough@phc.ox.ac.uk.].

Reviewers' comments:

Reviewer's Responses to Questions

**Comments to the Author**

1. Is the manuscript technically sound, and do the data support the conclusions?

Reviewer #1: Partly

Reviewer #2: Partly

2. Has the statistical analysis been performed appropriately and rigorously? 

Reviewer #1: No

Reviewer #2: No

3. Have the authors made all data underlying the findings in their manuscript fully available?

Reviewer #1: No

Reviewer #2: No

4. Is the manuscript presented in an intelligible fashion and written in standard English?

Reviewer #1: Yes

Reviewer #2: Yes

5. Review Comments to the Author

Reviewer #1: Overview

This paper uses interrupted time series analyses to evaluate the effects of the UK soft drink industry levy on sugar levels, price, volume, and non-sugar sweeteners. This is an important topic, and the authors provide strong justification for their study, which examines medium-term impacts of the intervention. The paper is interesting, well written and thorough. I have some concerns about the specification of the interrupted time series models, and have made some suggestions for the authors to consider and respond to. In particular, I suggest that the 6-monthly breakpoints in all models should be further justified or removed.

Comments

Abstract and Introduction: These sections are clear, with good justification provided for the study.

Outcome Measures: The definitions for time series would benefit from additional detail. For all measures, how was SKU-level information combined? My best guess is that it is the proportion of soft drink SKUs, and the average across soft drink SKUs (with each SKU weighted equally regardless of volume).

Statistical Methods: I am not fully convinced by the model specification of splitting the data into six month periods. I think this is essentially testing for a series of arbitrary breakpoints that are unlikely to best fit the series. From the figures, they do not look like they fit the data well. Are there any statistical papers recommending this approach?

In my view the specification should reflect the expected structure of the series – which is more likely to be a step and a slope change at the intervention point only. If the authors expect that the intervention effect will have decayed by the end of the series, and do not want late datapoints to influence the intervention effects, they might consider using a pulse effect instead of a slope change, perhaps as a sensitivity analysis e.g. https://www.ncbi.nlm.nih.gov/books/NBK563546/. If breakpoints are identified on visual inspection of the series, I would suggest including them, but only if they are significant and improve model parsimony.

Additional methods queries:

1. I wondered how changes to the Tesco clubcard scheme might have affected prices e.g. the launch of clubcard plus and clubcard prices in 2019. The authors may want to consider controlling for these changes.

2. The authors discuss the effect of inflation on the real-term value of the levy over time, and conclude that its effects have not diminished. I think this might be incorrect, as inflation means that similar price levels being maintained means soft drinks have become cheaper over time in real-terms. Perhaps the price analyses should use inflation-adjusted prices?

3. I wondered how lag 7 was chosen for auto-correlation adjustment, and whether any tests were used to confirm that this was sufficient.

Table 1: I am not highly familiar with the Wilcoxon rank sum test, but some of the significant p-values in Table 1 are confusing to me. For example, in the first row, exempt goes from 8.4 to also 8.4 which has an associated p-value of 0.031. Could the authors explain this?

Results: It would be very helpful to include a table that summarises all model results for step and slope changes (B, 95% CI, p-value). This could possibly focus on just the effects at the time of the intervention, or could include all effects if the authors apply my suggestions regarding model specifications.

Reviewer #2: Thank you for giving me a chance to the submitted manuscript.

Bandy and colleagues examined the midterm impact of the soft drink industry levy (SDIL) on sugar levels, price, portion size and use of non-sugar sweeteners in UK; the data source was obtained from leading retail supermarket and the primary analyses were done by interrupted time series (ITT). The authors found that reductions in sugar levels were sustained by 2020 and a reduction in product volume for higher tier drinks may be diminishing over time.

The manuscript is overall well written, but it has a number of points to be clarified as well.

Major points

The analysis specifically excluded considering slope changes between the various post-implementation time periods (in Statistical Methods section). However, it is noted that these slope changes were subsequently reported, leading to potential confusion. Please explain.

Minor points

1. In my understanding, the unit of analysis in this study was the number of available products (not the sales!) at the online supermarket. If this is correct, it would be more appropriate to mention this information earlier in the paper, rather than introducing it for the first time in the limitations section.

2. Abstract: “There was a weak evidence”: the strength of evidence does not depend on p-value alone.

3. Introduction: Do UK children admit for tooth extraction?

4. Methods; The authors have classified diet sodas as eligible drinks. However, it is unclear how the sugar levels in these drinks were determined. As a related issue, conducting sensitivity analyses that reclassify such drinks as non-eligible may be needed.

5. Methods; I am wondering why the authors considered the product volume and price as being for the total product in the case of multipacks."

6. Methods: I am unclear about the statement, 'it was imputed with the nearest available value for the same product before the missing value.' Could you please clarify if the data were imputed using values from the same product?"

7. Figure 1: Hard to visualize.

8. Reference: The reference style does not meet the Journal standards.

9. Reference: If available, please cite the URL (eg, TESCO 2023).

10. The manuscript is somewhat lengthy. I think some parts should be move to Supplemental information, including (but not limited to) “Display of results” or “Changes to Protocol”.

6. PLOS authors have the option to publish the peer review history of their article (what does this mean?). If published, this will include your full peer review and any attached files.

Reviewer #1: No

Reviewer #2: No

---

## [Author Response · Author response to Decision Letter 0]

29 Feb 2024

Please see attached response to reviewers document for full response

---

## [Decision Letter · Decision Letter 1]

26 Mar 2024

The impact of the UK Soft Drink Industry Levy on the soft drink marketplace, 2017 – 2020: an interrupted time series analysis with comparator series

PONE-D-23-34193R1

Dear Dr. Bandy,

We’re pleased to inform you that your manuscript has been judged scientifically suitable for publication and will be formally accepted for publication once it meets all outstanding technical requirements.

Kind regards,

Hans-Peter Kubis, PD. Dr. rer. nat.

Academic Editor

PLOS ONE

Additional Editor Comments (optional):

Reviewers' comments:

Reviewer's Responses to Questions

**Comments to the Author**

1. If the authors have adequately addressed your comments raised in a previous round of review and you feel that this manuscript is now acceptable for publication, you may indicate that here to bypass the “Comments to the Author” section, enter your conflict of interest statement in the “Confidential to Editor” section, and submit your "Accept" recommendation.

Reviewer #1: All comments have been addressed

Reviewer #2: All comments have been addressed

2. Is the manuscript technically sound, and do the data support the conclusions?

Reviewer #1: Yes

Reviewer #2: Partly

3. Has the statistical analysis been performed appropriately and rigorously? 

Reviewer #1: Yes

Reviewer #2: Yes

4. Have the authors made all data underlying the findings in their manuscript fully available?

Reviewer #1: No

Reviewer #2: No

5. Is the manuscript presented in an intelligible fashion and written in standard English?

Reviewer #1: Yes

Reviewer #2: Yes

6. Review Comments to the Author

Reviewer #1: The authors have provided thoughtful responses to my comments – thank you.

Statistical methods: I felt that the long response on model specification (re breakpoints) was very helpful. Could this potentially be included as a methodological supplementary file with the paper? I think it would be useful to readers who are interested in the methods.

Reviewer #2: The authors made the best efforts to address the concerns raised by the reviewers. I think this version is ready for the publication. Thank you.

7. PLOS authors have the option to publish the peer review history of their article (what does this mean?). If published, this will include your full peer review and any attached files.

Reviewer #1: No

Reviewer #2: No

---

## [Editor Report · Acceptance letter]

27 May 2024

PONE-D-23-34193R1 

PLOS ONE

Dear Dr. Bandy, 

I'm pleased to inform you that your manuscript has been deemed suitable for publication in PLOS ONE. Congratulations! Your manuscript is now being handed over to our production team.

Kind regards, 

on behalf of

Dr. Hans-Peter Kubis 

Academic Editor

PLOS ONE